# High Levels of MFG-E8 Confer a Good Prognosis in Prostate and Renal Cancer Patients

**DOI:** 10.3390/cancers14112790

**Published:** 2022-06-04

**Authors:** Karen Geoffroy, Patrick Laplante, Sylvie Clairefond, Feryel Azzi, Dominique Trudel, Jean-Baptiste Lattouf, John Stagg, Fred Saad, Anne-Marie Mes-Masson, Marie-Claude Bourgeois-Daigneault, Jean-François Cailhier

**Affiliations:** 1Institut du Cancer de Montréal (ICM), Centre de Recherche du Centre Hospitalier de l’Université de Montréal (CRCHUM), Montreal, QC H2X 0A9, Canada; karen.geoffroy@umontreal.ca (K.G.); laplantepat@videotron.ca (P.L.); sylvie.clairefond@usask.ca (S.C.); feryel.azzi.chum@ssss.gouv.qc.ca (F.A.); dominique.trudel.chum@ssss.gouv.qc.ca (D.T.); jean-baptiste.lattouf@umontreal.ca (J.-B.L.); john.stagg@umontreal.ca (J.S.); fred.saad@umontreal.ca (F.S.); anne-marie.mes-masson@umontreal.ca (A.-M.M.-M.); marie-claude.bourgeois-daigneault@umontreal.ca (M.-C.B.-D.); 2Division of Pathology and Cellular Biology, Université de Montréal, Montreal, QC H3C 3J7, Canada; 3Division of Urology, Department of Surgery, Université de Montréal, Montreal, QC H3C 3J7, Canada; 4Faculté de Pharmacie, Université de Montréal, Montreal, QC H3C 3J7, Canada; 5Department of Medicine, Faculté de Médecine, Université de Montréal, Montreal, QC H3C 3J7, Canada; 6Department de Microbiologie, Infectiologie et Immunologie, Faculté de Médecine, Université de Montréal, Montreal, QC H3C 3J7, Canada; 7Division of Nephrology, Department of Medicine, Université de Montréal, Montreal, QC H3C 3J7, Canada

**Keywords:** MFG-E8, prostate cancer, renal cancer, M2 macrophage

## Abstract

**Simple Summary:**

In the present study, we analyzed the distribution and prognostic impact of milk fat globule-epidermal growth factor-8 (MFG-E8) protein expression in patients with prostate and renal cancers. Our data highlighted MFG-E8 expression by tumor cells in the epithelium. Our results also showed that low levels of MFG-E8 in prostate and renal cancers were associated with worse clinical outcomes. Furthermore, higher numbers of CD206^+^ cells were found in the peripheral regions of renal clear cell carcinoma that expressed lower MFG-E8 levels. Globally, our results suggest that MFG-E8 expression could potentially be used as a prognostic marker in prostate and renal cancers.

**Abstract:**

Milk fat globule-epidermal growth factor-8 (MFG-E8) is a glycoprotein secreted by different cell types, including apoptotic cells and activated macrophages. MFG-E8 is highly expressed in a variety of cancers and is classically associated with tumor growth and poor patient prognosis through reprogramming of macrophages into the pro-tumoral/pro-angiogenic M2 phenotype. To date, correlations between levels of MFG-E8 and patient survival in prostate and renal cancers remain unclear. Here, we quantified MFG-E8 and CD68/CD206 expression by immunofluorescence staining in tissue microarrays constructed from renal (*n* = 190) and prostate (*n* = 274) cancer patient specimens. Percentages of MFG-E8-positive surface area were assessed in each patient core and Kaplan–Meier analyses were performed accordingly. We found that MFG-E8 was expressed more abundantly in malignant regions of prostate tissue and papillary renal cell carcinoma but was also increased in the normal adjacent regions in clear cell renal carcinoma. In addition, M2 tumor-associated macrophage staining was increased in the normal adjacent tissues compared to the malignant areas in renal cancer patients. Overall, high tissue expression of MFG-E8 was associated with less disease progression and better survival in prostate and renal cancer patients. Our observations provide new insights into tumoral MFG-E8 content and macrophage reprogramming in cancer.

## 1. Introduction

Milk fat globule-epidermal growth factor-8 (MFG-E8), also known as lactadherin, is a glycoprotein of 46 kDa found in many mammalian species, including mice and humans [1]. MFG-E8 is secreted as a soluble or microvesicular/exosomal protein by endothelial cells, epithelial cells, activated phagocytes, and apoptotic cells [2,3,4,5,6]. It contains two epidermal growth factor (EGF)-like domains in the N-terminal, one of which encodes an arginine-glycine-aspartate (RGD) motif that binds to α_V_β_3/5_ integrins, as well as two discoidin-like domains in the C-terminal that bind to phosphatidylserines (PS) on apoptotic cells [2,7]. This conformation allows MFG-E8 to act as an opsonin to promote phagocytosis of apoptotic cells and to regulate angiogenesis through interactions with the VEGF signaling pathway [8,9]. Notably, we and others have shown that MFG-E8 is involved with the regulation of the production of autoantibodies and wound healing and is implicated in different clinical conditions such as Alzheimer’s disease, atherosclerosis, sepsis, and unilateral ureteral obstruction [10,11,12,13,14,15,16].

MFG-E8 has also been recognized as a prognostic marker for cancer. MFG-E8 is overexpressed in different types of cancer such as melanoma [17], oral squamous cell carcinoma [18], and glioma [19], as well as breast [7,20], colorectal [21], esophageal [22], ovarian [23], pancreatic [24], bladder [25], and prostate [26] cancers. Except for one study in a specific subgroup of breast cancers, increased detection of MFG-E8 in malignant tissues was associated with poor outcomes for cancer patients. Multiple players have been identified as promoters of malignancy in response to MFG-E8, including endothelial cells and pericytes that play a crucial role in angiogenesis, and tumor-associated macrophages (TAMs), which are often considered as pro-tumoral macrophages [27].

TAMs are important players in the tumor immune environment and can participate in both tumor clearance and tumor progression [28,29]. Dynamic changes in TAM phenotypes are induced by the local microenvironment [30,31,32,33,34]. Pro-inflammatory/anti-tumoral macrophages (called M1) result from classical and innate activation triggered mostly by bacterial products. M1 express specific markers such as CD80/86 and the major histocompatibility complex class II (MHC-II) [35,36]. Anti-inflammatory/pro-tumoral macrophages (called M2) result from alternative activation that occurs when IL-4 or IL-13 are present [35,36]. This phenotype is characterized by the expression of specific markers such as CD206 and Arginase-1 [35,36]. Importantly, TAMs found in malignant tissues usually display an M2 phenotype [27,37], which impairs the immune-mediated tumor clearance. Prostate and renal cancers are often treated using various forms of immunomodulating therapies that could also modulate TAM functions [38,39]. Recently, analarmins such as MFG-E8 have been implicated in the generation of this immunosuppressive M2 phenotype [2,14]. Since numerous cancers express high levels of MFG-E8, its release may have an important role in macrophage reprogramming and tumor progression.

In the present study, we investigated MFG-E8 expression in tissue microarray (TMA) samples from renal and prostate cancer patients, two cohorts for which the association between MFG-E8 expression and patient outcomes remains unexplored. We evaluated the correlation between protein expression and patient progression and survival. Surprisingly, we found that prostate and renal cancer patients with high levels of MFG-E8 have a better prognosis. Furthermore, we found more CD206^+^ cells in the normal adjacent tissue of renal cancer than in malignant regions. Our study is the first to investigate the role of MFG-E8 in renal cancer and to report positive correlations between its expression and patient outcomes for both renal and prostate cancers.

## 2. Materials and Methods

### 2.1. Tissue Microarrays (TMAs)

The prostate cancer TMA was constructed as described previously [40]. The TMA contained specimens from 274 patients with prostate adenocarcinomas and 274 matched controls. Stratification by Gleason score resulted in 133, 92, 19, and 28 patients classified as 3 + 3, 3 + 4, 4 + 3, and 4 + 4, respectively. Clinicopathological characteristics of patients with prostate cancer are provided in Table 1.

For the renal TMA, samples were collected and banked following appropriate consent from 190 patients undergoing kidney surgery at the Centre hospitalier de l’Université de Montréal (CHUM) from 2007 to 2015. The selected patients had no pre-operative chemotherapeutic treatment. Clear cell, papillary, or chromophobe carcinoma subtypes were included in the TMA as follows: 142 patients with clear cell renal carcinoma (CCRC), 34 with papillary renal cell carcinoma (PRCC), and 14 with chromophobe renal cell carcinoma (CRCC). Malignant and nonmalignant matching tissues were cored in duplicates using a 0.6 mm diameter tissue TMArrayer (Pathology Device Inc., San Diego, CA, USA) and resultant cores were randomly arranged into three paraffin blocks. Distance between tumor and stromal tissue was less than 500 μm for each specimen. The TMAs were then sectioned (4 μm), stained with hematoxylin–eosin, and the presence or absence of tumor tissue was confirmed by a pathologist. According to the WHO/ISUP grading system, 1 patient was grade 1, 53 patients were grade 2, 89 patients were grade 3, and 32 patients were grade 4. Clinicopathological characteristics of patients with renal cancer are provided in Table 2. 

### 2.2. Immunofluorescence (IF)

For both TMAs, antigen retrieval was carried out using the Cell Conditioning #1 solution for 60 min with the Benchmark XT autostainer (Ventana Medical System Roche, Oro Valley, AZ, USA). IF staining was used to quantify MFG-E8 expression and to delineate the epithelium. Slides were incubated for 60 min with monoclonal mouse anti-human MFG-E8 (diluted 1/250, Santa-Cruz Biotechnology #SC-8029, Dallas, TX, USA) and monoclonal rabbit anti-human CK 8/18 (1/100, Agilent #MA514428, Santa Clara, CA, USA). The slides were washed in PBS and incubated for 20 min with the protein-block from Agilent. Slides were incubated for 60 min in 1% BSA/PBS with donkey anti-mouse-Cyanine5 (1/250, ThermoFisher Scientific #A31571, Waltham, MA, USA) and goat anti-rabbit Alexa Fluor 546 (1/250, ThermoFisher Scientific #A11010), and washed again. Nuclear localization was performed using DAPI counterstain solution (1/1000, ThermoFisher Scientific #D1306) for 10 min. Slides were washed three times and incubated for 15 min with Sudan Black B solution (1/1000, Sigma-Aldrich, Oakville, ON, Canada). After a final wash, slides were mounted with Fluoromount Aqueous Mounting Medium (Sigma Aldrich), stored at 4 °C, and scanned the following day with Olympus Optical Microscope model BX61VSF (Olympus, Shinjuku, Tokyo, Japan). Similarly for macrophage phenotypes, we used monoclonal mouse anti-human CD68 (1/500, Agilent #M0876), polyclonal rabbit anti-human CD206 (1/1000, Abcam #AB64693, Cambridge, UK), and monoclonal mouse anti-human CK7/19 and CK18 (1/200 each, ThermoFisher Scientific #MA511986 and Santa-Cruz Biotechnology #SC-376126, respectively). We blocked TMA slides with Mouse-On-Mouse (MOM) Blocking Reagent (4 drops in 1 mL PBS, Vector Laboratories, Burlingame, CA, USA) overnight at 4 °C before adding secondary antibodies for 1 h (1/200 each): Alexa Fluor 488 goat anti-mouse, Cy5 goat anti-mouse, and TRITC goat anti-rabbit, all from ThermoFisher Scientific (#A11001, #A10524, and #A16101, respectively). Negative control (omission of primary antibodies) TMA slides were stained in parallel for each experiment and fluorescent signals obtained were used as thresholds.

Quantification of MFG-E8 expression and CD68/CD206 was performed using Visiopharm Integrator System software (VIS; Visiopharm, Denmark), allowing for automated image analysis. Analysis protocols were designed regardless of the histopathological grade. Surface percentage of MFG-E8-positive cores was assessed by calculating the ratio between MFG-E8-positive area and total sample area.

### 2.3. Statistical Analysis

Only cores in which more than 5% of epithelial cells were detected and with a total area greater than 100,000 µm^2^ were included in our analysis. Results from different groups were compared using Mann–Whitney test or ANOVA from GraphPad Prism software (V6, GraphPad, La Jolla, CA, USA) as indicated. Statistical survival and disease progression analyses were performed using the SPSS Statistics software (v25.0, SPSS Inc., Chicago, IL, USA). Survival curves were generated using the Kaplan–Meier method, and the log-rank test was used to evaluate significant differences. A *p*-value < 0.05 was considered statistically significant.

## 3. Results

### 3.1. Localization of MFG-E8 Expression in Malignant Areas of Prostate and Renal Cancer Tissues

Using TMAs containing specimens from 274 prostate adenocarcinoma patients (Table 1) and 190 renal carcinoma patients (Table 2), we examined intracellular MFG-E8 protein expression in malignant tissue by IF microscopy (Figure 1). Our results showed that MFG-E8 was expressed at different levels by tumor cells (expressing cytokeratin 8/18) with regards to malignancy and type of cancer. When we compared MFG-E8 expression in malignant versus normal adjacent tissues, defined as ‘benign’ for prostate specimens and ‘peripheral’ for renal specimens, we observed statistical differences according to tumor type. For prostate cancer and PRCC (papillary renal cell carcinoma), MFG-E8 expression was higher in malignant areas compared to nonmalignant areas (Figure 2A,C). In contrast, MFG-E8 was expressed at lower levels in malignant areas compared to peripheral areas in CCRC (clear cell renal carcinoma) samples (Figure 2B). No differences were observed in the CRCC (chromophobe renal cell carcinoma) for which few specimens were available for analysis (data not shown). These results suggest that MFG-E8 protein expression differs between malignant and nonmalignant areas in prostate and renal cancers.

### 3.2. MFG-E8 Expression Differs According to the Grade of Cancer

We then investigated whether MFG-E8 expression correlated with cancer grade. We first observed that MFG-E8 expression in benign areas was independent of grade for prostate cancer (Figure 3A) and peripheral areas for renal cancer (data not shown). Interestingly, MFG-E8 expression in the malignant zones of prostate cancer (Figure 3A) and CCRC (Figure 3B) decreased significantly as the cancer grade increased. No statistical differences were observed in the PRCC subgroup (Figure 3C).

### 3.3. CD206^+^ Cells Are Differentially Distributed in Renal Cancers

Given that MFG-E8 affects the macrophage phenotype, we evaluated the presence of CD68^+^/CD206^+^ and CD68^−^/CD206^+^ cells in the malignant and peripheral regions of CCRC and PRCC (Figure 4A). CD206^+^ cells were significantly increased in the peripheral normal adjacent zones of these two renal cancers (Figure 4B,C), whereas no differences were observed in prostate cancer (data not shown). These results suggest a potential association between low levels of MFG-E8 in the malignant areas and the increased presence of M2-like TAMs in peripheral normal adjacent areas. However, there was no association between MFG-E8 and CD206^+^ cells.

### 3.4. MFG-E8 Expression Correlates with Patient Outcome

We evaluated the prognostic impact of MFG-E8 protein expression (described as “high” or “low” from their median MFG-E8 expression) in malignant tissue by Kaplan–Meier analyses. Surprisingly, we observed that low MFG-E8 protein expression was significantly associated with a worse prognosis for both prostate and renal cancer patients (Figure 5). Prostate cancer patients were significantly more susceptible to biochemical recurrence (BCR) and bone metastasis (BM) development when MFG-E8 was expressed at lower levels. Our analyses failed to highlight a significant relation to overall survival (OS) (Figure 5A, right panel). Similarly, in renal cancer patients, low expression of MFG-E8 correlated significantly with a decreased progression-free survival (PFS) and OS (Figure 5B). Taken together, our results demonstrate that MFG-E8 expression correlates with the outcome of patients with prostate or renal cancer.

## 4. Discussion

The objective of this study was to evaluate the distribution and prognostic impact of MFG-E8 protein expression in patients with prostate and renal cancers. Our results showed that MFG-E8 was expressed by tumor cells in the epithelium, predominantly in malignant areas for both prostate and PRCC. Interestingly, we observed that MFG-E8 levels were reduced in malignant areas within the CCRC subgroup when compared to nonmalignant areas. In addition, we observed that MFG-E8 levels were significantly reduced in malignant areas as the cancer grade increased. Moreover, CD206^+^ cells were more distributed in the peripheral normal adjacent zone in the tissues of renal cancer patients. Finally, our results showed that low levels of MFG-E8 in prostate and renal cancers were associated with worse clinical outcomes.

To our knowledge, this is the first study discriminating MFG-E8 expression levels and patient survival in renal cancer. MFG-E8 has been well studied in kidney injury [12,41] and diabetic nephropathy [42], but its role in kidney cancer remains unexplored. Little is known for prostate cancer as well, except for the work of Soki et al., which showed an increased expression of MFG-E8 in the tissues and exosomes of prostate cancer patients [26]. Despite the low number of specimens present in their TMA, they found that positive MFG-E8 cells were in proximity to CD68^+^ cells, classically associated with macrophages. This result opened the discussion on the important role of TAMs, particularly M2 pro-tumoral macrophages in the progression of cancer. This is especially relevant with the fact that MFG-E8 can induce a phenotype transition from an M1 (anti-tumoral) to an M2 macrophage [14].

M2 TAMs are known to promote cancer progression through various mechanisms, including angiogenesis, extracellular matrix degradation, and tumor cell motility [43,44,45]. The mechanism involved in the polarization toward this pro-tumoral phenotype is still unclear, but the microenvironment present around cancer cells offers certain opportunities. First, tumors have a high rate of proliferation and apoptosis, and macrophages, also known as the phagocytic or scavenger cell, are rapidly recruited for the clearance of apoptotic cells (or efferocytosis), which promotes the acquisition of an M2 phenotype [26,46]. Second, apoptotic cells present in highly proliferative cancers release MFG-E8 that can directly, without efferocytosis, promote M2 macrophage reprogramming [2,12,14]. Moreover, multiple studies have detected high levels of MFG-E8 in tumors from different cancer types, which in turn, has the capacity to promote M2 polarization [9,14]. MFG-E8 also has the capacity to enhance efferocytosis by bridging apoptotic cells and the macrophage. Taken together, MFG-E8 represents a key factor from the tumor microenvironment that enhances M2 TAM reprogramming. In our renal TMA, we found a significant increase in M2 TAMs in peripheral normal adjacent areas. Low MFG-E8 expression in the malignant zones of CCRC associated with higher numbers of CD206^+^ cells in the periphery could suggest that the released MFG-E8 reprogrammed phagocytes into M2 TAMs after its diffusion into the adjacent tissue. However, no correlations with MFG-E8 levels were detected, suggesting in part that alternative mechanisms of macrophage differentiation M2 macrophages are also characterized by the secretion of various cytokines, chemokines, and growth factors, such as the VEGF, a well-known pro-angiogenic protein that has a profound impact on neovascularization and tumor progression [2,24]. Evidence has shown numerous cross-talks between MFG-E8 and VEGF signaling pathways, especially with the association of α_v_β_3_ and VEGF Receptor-2 [47]. However, the importance of MFG-E8 in the angiogenic-dependent growth of tumors is still under debate [7,9,24,25]. The relationship between these two proteins still requires evaluation in our TMAs, in which macrophages represent a key player in the tumor microenvironment, as observed in our renal cohort.

Although we show that high levels of MFG-E8 correlate to better patient outcomes in prostate and renal cancer patients, our results are in opposition with most of the literature, which associates high expression of MFG-E8 with bad prognoses in cancer patients [25,48,49]. Only the study of Yang et al. showed that low expression of MFG-E8 in invasive lesions was associated with tumor progression in ER^+^/ERBB2^+^ breast carcinoma patients [50]. In colorectal cancer, MFG-E8 was involved in the growth and migration of tumors through an epithelial-to-mesenchymal transition (EMT) mechanism [21]. In skin cancer, MFG-E8 expression also favored EMT, invasion and resistance to apoptosis of cancer cells, as well as upregulation of the Treg population, all of which are pro-tumoral mechanisms [9,17]. In bladder cancer, reduction in tumor growth in MFG-E8-deficient mice was compromised in mice lacking functional T and B cells, suggesting a role for the immune system [25]. For all these processes induced by MFG-E8, multiple signaling pathways have been identified as important in tumor progression, such as the α_v_β_3/5_ integrin, the PI3K/Akt, and the STAT3/SOCS3 pathways. To integrate all these hypotheses, we propose a model on the possible roles of MFG-E8 in the cancer microenvironment from our work and others (Figure 6). This model integrates our findings and proposes that apoptotic renal epithelial cancer cells release MFG-E8, which induces an M2 macrophage phenotype, promoting a favorable milieu to the tumor. Other potential pro-cancerous mechanisms (neoangiogenesis, EMT, treatment resistance) driven by released MFG-E8, either soluble or by microvesicles/exosomes, could also be detrimental to patients.

One limitation of our study is the depth in the conclusions that can be extrapolated from a correlation study. For example, each patient sample used in our TMAs for multi-color IF microscopy was obtained at a given time, and so, patient/sample status overtime was not assessed. Another important limitation is that MFG-E8 is a secreted protein and can be difficult to detect by IF because it diffuses away from the site of production instead of accumulating in each cell/microenvironment. Our analysis therefore focused on the signal detected inside cells, but it may not properly reflect expression levels. In addition, our IF stains were performed on two successive slides. Given the high heterogeneity observed in cancer, it remains possible that different sections from the same samples would lead to different conclusions. Because of resource limitations, we could not perform more IF studies. Finally, the cohort used for Figure 3C is smaller compared to the others. Although we did observe a trend in our analysis, the results are not statistically significant. A bigger cohort would have strengthened our conclusions.

## 5. Conclusions

Taken together, our results suggest that MFG-E8 expression could potentially be used as a prognostic marker in prostate and renal cancers. The role of MFG-E8 in the tumor microenvironment has led to strategies that reduce the pro-tumoral potential of MFG-E8 using blocking antibodies to neutralize its interactions with cell receptors [23]. However, our results raise more questions into the mechanisms behind low MFG-E8 levels and require further investigations. Future studies should determine whether low MFG-E8 expression is due to low transcription/translation or is reflective of increased secretion with pro-tumoral paracrine consequences on endothelial cells/pericytes and macrophages in the normal adjacent peripheral zones. These investigations will provide further insight into how the addition of MFG-E8 or activation of its signaling pathways may confer an advantage in some cancers.

## Figures and Tables

**Figure 1 cancers-14-02790-f001:**
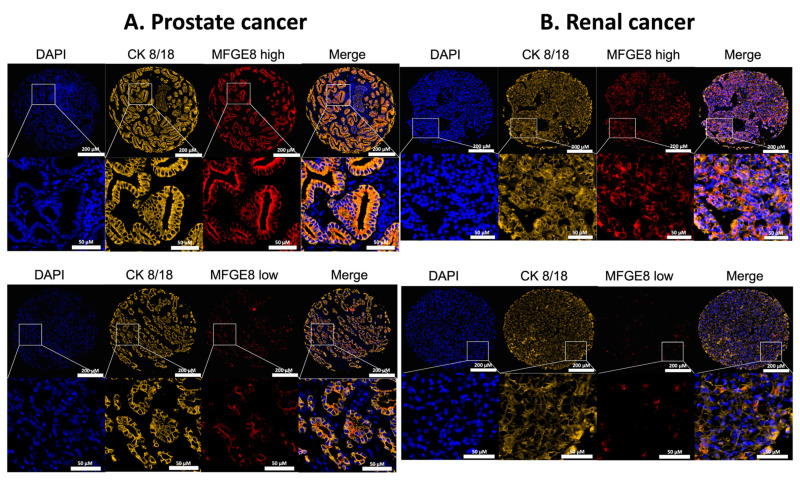
Expression of MFG-E8 in malignant areas of prostate and renal cancer tissues. Representative images of IF staining in prostate (**A**) and renal (**B**) cancer TMAs, grade 3 + 4 and grade 3, respectively. DAPI (blue) = nuclei, CK 8/18 (yellow) = epithelial cells, and MFG-E8 (red). The white boxes represent zoomed-in areas. Scale bars: 200 μm for whole image of sample, 50 µm for zoomed-in images. Images represent examples of high (top panels) and low (bottom panels) levels of MFG-E8 expression for both types of cancer.

**Figure 2 cancers-14-02790-f002:**
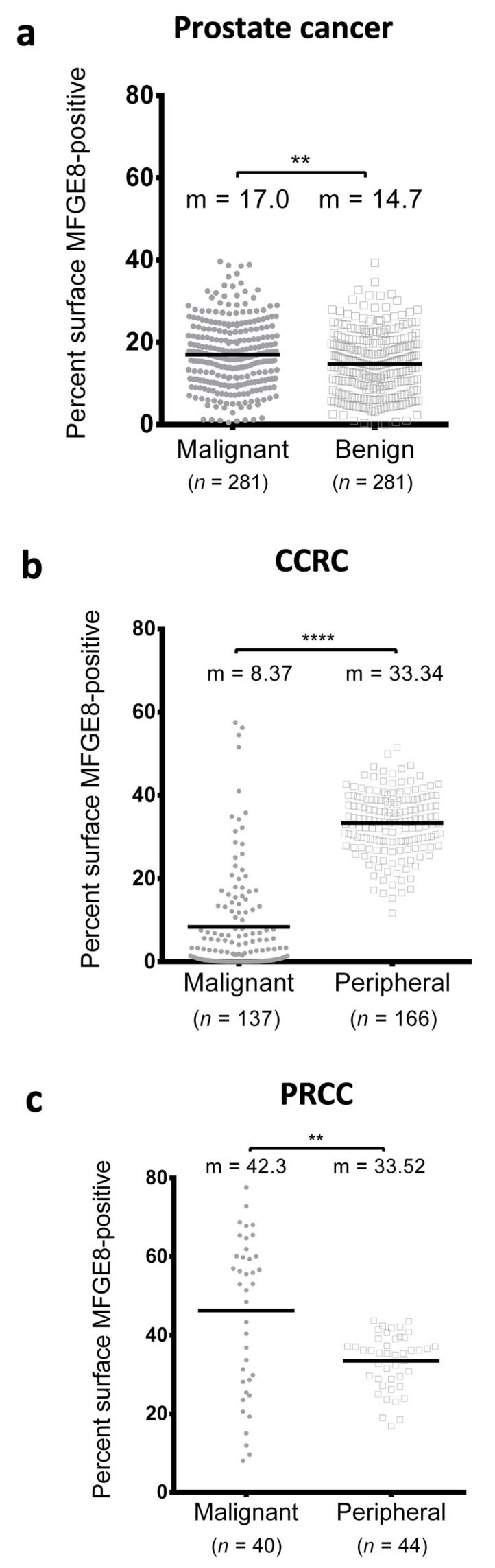
Percentage of MFG-E8-positive surface in malignant and benign areas for prostate and renal cancer tissues. We compared the percentage of MFG-E8-positive staining in prostate cancer (**a**), CCRC (**b**), and PRCC (**c**) specimens in malignant (grey dots) and nonmalignant areas (benign/peripheral = white squares). Statistical analyses were performed using Mann–Whitney test: m = mean; ** *p* ≤ 0.01; **** *p* ≤ 0.0001.

**Figure 3 cancers-14-02790-f003:**
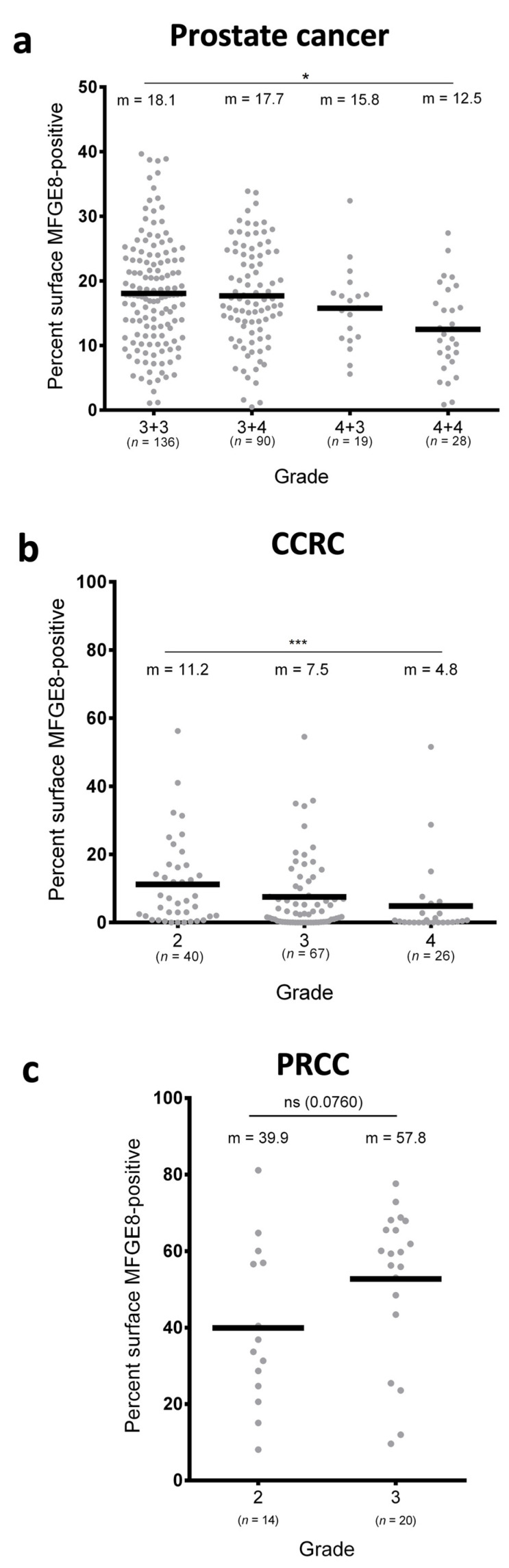
Expression of MFG-E8 according to prostate and renal cancer grades.Comparison of the percentage of MFG-E8-positive staining between prostate cancer (**a**), CCRC (**b**), and PRCC (**c**) specimens with regards to cancer grade and malignancy. Statistical analyses were performed using one-way ANOVA test: m = mean; ns = not significant; * *p* ≤ 0.05 (3 + 3 vs. 4 + 4); *** *p* ≤ 0.001 (2 vs. 4).

**Figure 4 cancers-14-02790-f004:**
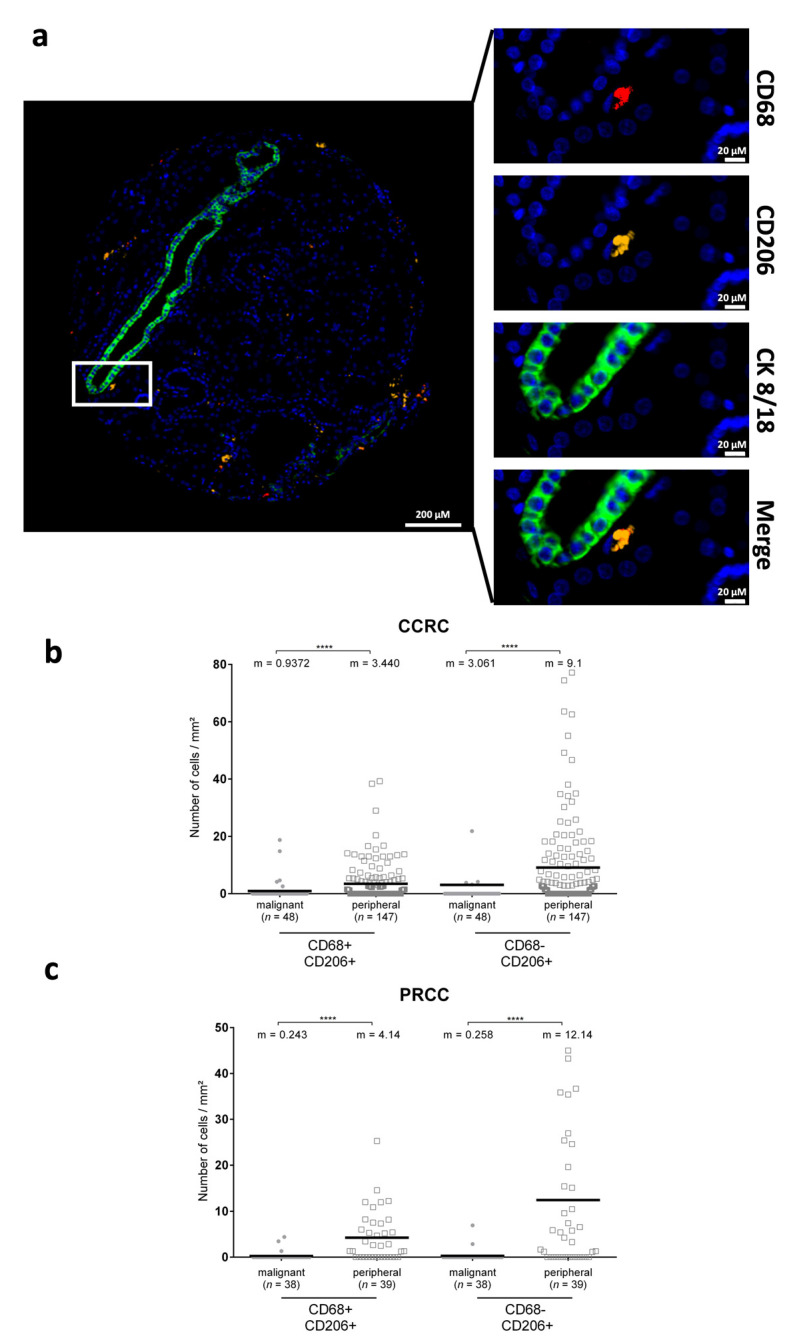
M2 TAMs in renal cancer tissues. (**a**) Representative images of IF staining for CD68 and CD206 in renal cancer TMAs. DAPI (blue) = nuclei, CD68 (red), CD206 (yellow), and CK8/18 (green). Image panel on the right represents a zoom-in of the white box in the left image. Bar = 20 µm. CD68^+^/CD206^+^ cells (M2 macrophages) and CD68^−^/CD206^+^ cells were decreased in malignant areas (grey dots) compared with nonmalignant areas (white squares) in both CCRC (**b**) and PRCC (**c**). Statistical analyses were performed using two-sided Mann–Whitney test: m = mean; **** *p* ≤ 0.0001.

**Figure 5 cancers-14-02790-f005:**
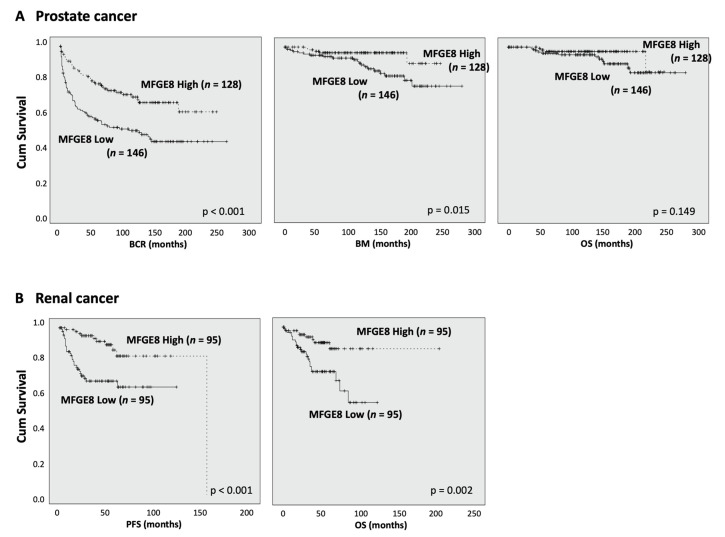
Patient survival according to MFG-E8 expression in prostate and renal cancer tissues. Kaplan–Meier analyses were performed for prostate (**A**) and renal (**B**) cancer patients based on the median value of MFG-E8 expression (classified as high (17%) and low (7.6%)). For prostate cancer, endpoints for analyses included biochemical recurrence (BCR), bone metastasis (BM), and overall survival (OS). For renal cancer, endpoints included progression-free survival (PFS) and OS. A value of *p* ≤ 0.05 was considered statistically significant.

**Figure 6 cancers-14-02790-f006:**
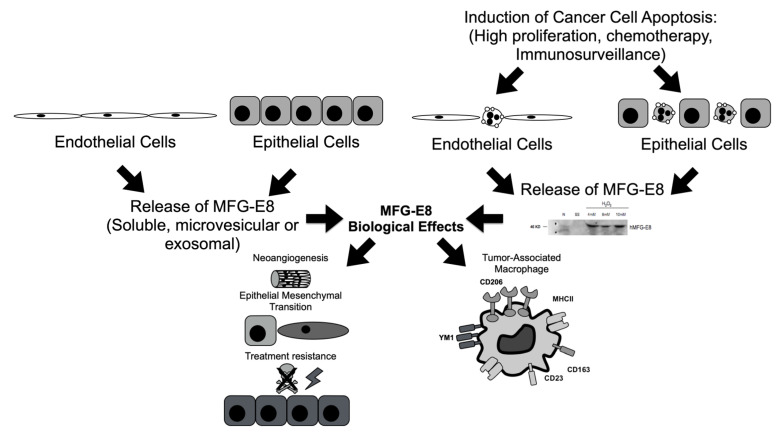
Schematic diagram of our hypothesis. MFG-E8 is released as a soluble or vesicular protein by endothelial and epithelial cells within the tumor microenvironment. MFG-E8 exerts its pro-tumoral effects by promoting neoangiogenesis, epithelial-mesenchymal transition, and treatment resistance. In parallel, apoptotic endothelial (as we published) and renal epithelial cancer cells secrete MFG-E8 (as shown in the Western blot), and through various mechanisms discussed above, will favor M2 macrophage reprogramming or tumor-associated macrophages. Altogether, these mechanisms lead to a pro-tumoral microenvironment that is deleterious to cancer patients.

**Table 1 cancers-14-02790-t001:** Characteristics of the prostate cancer patient cohort. Of 274 patients, 146 were classified as MFG-E8-high and 128 as MFG-E8-low. Prostate-specific antigen (PSA) had a median concentration of >7.1 ng/mL and >6.9 ng/mL (normal <4.0 ng/mL) in MFG-E8-low and MFG-E8-high patients, respectively. Positive margins were observed in 92/274 samples. Gleason scores: 1 = small uniform glands, 2 = more spaces between glands, 3 = distinct infiltration of cells from glands at margins, 4 = irregular masses of neoplastic cells with few glands, 5 = lack of glands, sheets of cells. Follow-up time: the number of months between the date of primary resection of tumor and last contact with the patient. Progression time: the number of months at which bone metastases (BM) or biochemical recurrence (BCR) occurred following surgery. cTNM = clinical Tumor-Node-Metastasis, pTNM = postoperative Tumor-Node-Metastasis, LNI = Lymph Node Invasion, SVI = Seminal Vesicle Invasion and CRPC = Castration-Resistant Prostate Cancer.

Prostate Cancer Cohort	Characteristics	MFG-E8 Low	MFG-E8 High
Number of patients		146	128
Diagnosis	Median age (years)	63 (51–73)	63 (47–74)
PSA pre-surgery	Median rate (ng/mL)	7.1 (1.6–37)	6.9 (2.4–40.5)
Margin		54	38
Gleason score	3 + 3	62	71
	3 + 4	48	42
	4 + 3	11	8
	4 + 4	21	7
cTNM	1	31	21
	2	115	107
pTNM	2	97	95
	3	44	29
	4	5	4
Follow-up		0–280	0–246
Progression time	BCR—Median (months)	74 (55)	37 (87)
	BM—Median (months)	20 (133)	5 (122)
Death rate	From prostatic cancer	12	4
	From all causes	26	20
Other	LNI	5	4
	SVI	8	8
	Extra-prostatic extension	48	30
	CRPC	21	4

**Table 2 cancers-14-02790-t002:** Characteristics of the renal cancer patient cohort. The 190 patients were first subdivided using expression levels of MFG-E8 (low vs. high) and then classified according to sex (men or women), antecedents (any cancer in the family), localization of tumor, histopathology (clear cell renal carcinoma (CCRC), papillary renal cell carcinoma (PRCC) or chromophobe renal cell carcinoma (CRCC)), stages (WHO/ISUP grading system), and surgery (complete or partial). Follow-up time: the number of months between the date of primary resection of tumor and last contact with the patient. Progression time: the number of months before which the cancer reoccurred following surgery. OS = overall survival and DSS = disease-specific survival.

Renal Cancer Cohort	Characteristics	MFG-E8-Low	MFG-E8-High
Number of patients		95	95
Sex	Women	28	24
	Men	67	71
Antecedents	Cancer in family	16	14
	Renal cancer	2	0
Laterality	Left	47	42
	Right	48	53
Histopathology	CCRC	78	64
	PRCC	10	24
	CRCC	7	7
Stages	1	0	1
	2	26	27
	3	45	42
	4	17	15
Follow-up		0–123	0–204
Surgery	Complete	51	44
	Partial	39	50
Progression	Median (months)	33	39
Survival	OS—Median (months)	42	42
Other	Capsular invasion	12	37
	Regional nodes	0	9
	Necrosis	28	66
	Pre-operative chemotherapy	4	7

## Data Availability

TMA data that support the findings of this study are available, but because of data which were used under license for the current study, they are not public and are available from the authors upon reasonable request.

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
