# Peer review of "High Levels of MFG-E8 Confer a Good Prognosis in Prostate and Renal Cancer Patients"

_cancers, 2022, doi:10.3390/cancers14112790_

Round 1
Reviewer 1 Report
In recent years, MFG-E8 has begun to emerge as a possible prognostic marker for several kinds of cancers. In the present clinical study, Geoffroy et al found that low levels of MEF-E8 were associated with poor outcomes in renal and prostate cancer subjects. The study is important and will provide lots of intellectual materials for researchers that are interested in mechanistic studies.
Strength of the study.
A sufficient number of subjects were taken into consideration. Inclusion and exclusion criteria are robust. The introduction provides sufficient details to understand the background and question being tackled in the study.
The quality of Figure 1 (IF) data is very good.
The authors have addressed all the possible limitations that reflect the honesty of the team.
I have only one concern about the study. The authors should reconsider the statistical analysis of Figure 2A.
Minor
The manuscript should be thoroughly checked for typographical errors.
Author Response
Reviewer 1:
Point 1: I have only one concern about the study. The authors should reconsider the statistical analysis of Figure 2A.
Response 1: Figure 2A represents the percentages of MFG-E8-positive surface in malignant and benign areas for prostate cancer tissues. We compared the percentage of MFG-E8-positive staining in prostate cancer. We can see ** in the figure that represents p ≤ 0.01 using the Mann-Whitney test. The high number of patients explain why the mean % is significant despite the presence of a small difference between the two groups.
Point 2: The manuscript should be thoroughly checked for typographical errors.
Response 2: The manuscript was checked for typographical errors.
Reviewer 2 Report
In the proposed study, Geoffroy and colleagues analyzed the distribution of MFG-E8 and CD68 / CD206 in tissue microarrays from prostate and kidney cancer. The data obtained were correlated with the available data from patient prognosis. This is a work that contains only statistical correlation data and therefore is largely speculative. The conclusions are based on hypotheses that have not been sufficiently tested experimentally but only on plausibility effects, therefore they are difficult to share, also because data were obtained on two very different tumor models.
Author Response
Reviewer 2:
Point 1: This is a work that contains only statistical correlation data and therefore is largely speculative. The conclusions are based on hypotheses that have not been sufficiently tested experimentally but only on plausibility effects, therefore they are difficult to share…
Response 1: Our work presents a powerful association of MFG-E8 expression with survival in prostate and renal cancer to suggest that it could be a new prognostic tool in these populations. That is our only conclusion. We put forward a hypothesis to explain how low intracellular content of MFG-E8 could reflect its release leading to diverse biological impacts such as macrophage reprogramming into M2 phagocytes and neoangiogenesis as depicted in our new diagram (Figure 6). To further strengthen this hypothesis, we now include new in vitro data on the release of MFG-E8 by apoptotic renal epithelial cells (see insert in Fig. 6). We already demonstrated that MFG-E8 induces M2 macrophage reprogramming (Laplante et al, JID 2017). Also, and as mentioned in our limitations, TMAs have limited possibilities with regards to biological readouts.
Point 2: …also because data were obtained on two very different tumor models.
Response 2: We think that confirmation of our associations in two different tumor models confers strength to our observation that MFG-E8 tumor cell content corelates with patient’s prognosis in prostate and kidney cancer.
Reviewer 3 Report
This is informative, correlative and descriptive research manuscript regarding the discovery of MFG-E8 production potentially impacting the development of pro-tumor macrophages known as M2 Tumor Associated Macrophages (TAM) in prostate and renal patients. Outstanding data, survival curves and immunoflorences images of MFG-E8 expression in prostate an renal tissues from patients. High MFG-E8 levels predict a good prognosis for patients with prostate or renal cancer. However, it would be ideal if this study could add more mechanistic and functional data in order to truly claim that MFG-E8 is a potential biomarker for treatment of prostate and renal cancers. For example, what regulates MFG-E8 production? Also if possible the authors can demonstrate in vitro that the reduction or absence of MFG-E8 production inturn cause protumor M2 Macrophages to reprogram into tumoricidal M1 Macrophages in prostate or renal tumor microenviroment? This data will provide strong supporting evidence. How does MFG-E8 potential regulate M1 vs. M2 Macrophage development? The author could provide a proposed model based on the current and past data reported which can strengthen this manuscript. The authors should carefully review their language regarding the polarization of M1 Macrophages into M2 Macrophages vs. M2 polarization into what? Also note that, M2 Macrophages can be reprogrammed into M1 Macrophages. The authors could further stained for more specific M2 macrophage markers beyond CD86 and CD206 by using YM1and Arginase and as well as specific M1 macrophage makers such as, MHC-II, NO and other ROS species from prostate and renal cancer patients bio-banked tissues for immunofluorences (IF) experiments.
Author Response
Reviewer 3
Point 1: However, it would be ideal if this study could add more mechanistic and functional data in order to truly claim that MFG-E8 is a potential biomarker for treatment of prostate and renal cancers.
Response 1: We do not claim that MFG-E8 is a biomarker for treatment. Based on our observations, we present it as a potential prognosis biomarker since patients with low MFG-E8 content had a worse outcome.
Point 2: For example, what regulates MFG-E8 production?
Response 2: We now include references on what is known on the secretion of MFG-E8 in the introduction. Text was also modified accordingly (Page 2, lines 56-57).
Point 3: Also if possible the authors can demonstrate in vitro that the reduction or absence of MFG-E8 production in turn cause protumor M2 Macrophages to reprogram into tumoricidal M1 Macrophages in prostate or renal tumor microenvironment? This data will provide strong supporting evidence.
Response 3: We put forward a hypothesis to explain how low intracellular content of MFG-E8 could reflect its release leading to diverse biological impacts such as macrophage reprogramming into M2 phagocytes and neoangiogenesis as depicted in our new diagram (Figure 6). We already demonstrated that MFG-E8 induces M2 macrophage reprogramming (Laplante et al, JID 2017).
Point 4: How does MFG-E8 potential regulate M1 vs. M2 Macrophage development? The author could provide a proposed model based on the current and past data reported which can strengthen this manuscript.
Response 4: We mentioned in the discussion various potential mechanisms by which MFG-E8 generates M2 Macrophages (page 12, lines 329-338 ). We also add a new diagram (Fig. 6) in the revised manuscript as a model proposition based on current and past data.
Point 5: The authors should carefully review their language regarding the polarization of M1 Macrophages into M2 Macrophages vs. M2 polarization into what? Also note that, M2 Macrophages can be reprogrammed into M1 Macrophages.
Response 5: We reviewed our language on polarization. We agree that M2 Macrophages can be reprogrammed into M1 Macrophages. We focussed mainly on the M2 phenotype since it is generally overrepresented in the tumor-associated macrophages (TAMs).
Point 6: The authors could further stain for more specific M2 macrophage markers beyond CD86 and CD206 by using YM1 and Arginase and as well as specific M1 macrophage makers such as, MHC-II, NO and other ROS species from prostate and renal cancer patients bio-banked tissues for immunofluorences (IF) experiments.
Response 6: We did not perform staining for other macrophages markers on our cohort of patients due to lack of labor and financial resources. However, as mentioned in Laplante et al 2017 (cited above and in the manuscript), MFG-E8-treated macrophages downregulated iNOS (M1 marker) and up-regulated EGR-2, CD206 and YM1 (M2 markers).
Reviewer 4 Report
MFG-E8(Milk fat globule-EGF factor 8), a secreted glycoprotein, plays an exceptional role in various diseases and solid tumors. In fact, MFG-E8 overexpression is found in a variety of cancers. However, it remains unclear whether MFG-E8 overexpression is associated with the clinicopathological characteristics and prognosis of human malignancies. Several papers have been published in the last decade regarding this topic.
Some changes are necessary.
- The background regarding MFG-E8 should be better discussed, as well as that regarding emerging treatment options in genitourinary malignancies. Some papers should be cited in my opinion (PMID: 30612778; PMID: 33714725; PMID: 32911806)
- figures and tables are very interesting, and the authors should be commended. However, figure 1 is quite blurry and should be corrected.
- A high number of samples, congratulations.
- the discussion should be expanded, including a more personal perspective.
Major changes are suggested.
Author Response
Reviewer 4
Point 1: The background regarding MFG-E8 should be better discussed, as well as that regarding emerging treatment options in genitourinary malignancies. Some papers should be cited in my opinion (PMID: 30612778; PMID: 33714725; PMID: 32911806)
Response 1: We modified the introduction with the proposed references.
Point 2: However, figure 1 is quite blurry and should be corrected.
Response 2: We apologize for that. We now include a revised figure 1 in the revised manuscript.
Point 3: …the discussion should be expanded, including a more personal perspective.
Response 3: We modified the discussion to reflect our personal perspective in the light of our new results as shown in our new figure (Fig. 6).
Round 2
Reviewer 2 Report
authors have not added significant changes in the revised version
Reviewer 3 Report
Authors addressed and completed the requested revisions.
Great proposed model for the work reported.
Please note minor issues with manuscript below:
Line 29 -proper spelling of prognosi?
50uM scale is not readable for Figure 1. and Figure 4.
Line 338- delete either Tumor associated macrophage or TAM
Reviewer 4 Report
The authors addressed all the issues we raised.
We recommend Acceptance.